# The magnitude of mental distress and associated factors among a school of medicine and college of health sciences students at Debre Markos University, 2021

Baye Tsegaye Amlak[1], Mezinew Sintayehu Bitew[2], Asmamaw Getnet[2], Fentahun Minwuyelet Yitayew[2], Tamene Fetene Terefe[1], Tadesse Tsehay Tarekegn[1], Asmare Getie Mihret[3], Omega Tolessa Geleta[2], Gebrie Getu Alemu[4], Fisha Alebel GebreEyesus[2], Dejen Tsegaye[2]*

1 Department of Nursing, College of Medicine and Health Sciences, Wolkite University, Wolkite, Ethiopia, 2 Department of Nursing, College of Health Sciences, Debre Markos University, Debre Markos, Ethiopia, 3 Department of Nursing, College of Medicine and Health Sciences, Arba Minch University, Arba Minch, Ethiopia, 4 Department of Nursing, Bahir Dar Health Science College, Bahir Dar, Ethiopia

* dejenetsegaye8@gmail.com

**Data Availability Statement:** All relevant data are within the paper and its Supporting information files.

## Abstract

### Introduction

Mental distress is a collection of mental health abnormalities characterized by symptoms of anxiety, depression, insomnia, fatigue, irritability, forgetfulness, difficulty in concentrating, and somatic symptoms. It affects society as a whole and no group is immune to mental distress; however, students have a significantly high level of mental distress than their community peers. The study is aimed to assess the magnitude of mental distress and associated factors among a school of medicine and college of health sciences students.

### Objective

To assess the magnitude of mental distress and associated factors among a school of medicine and college of health sciences students at Debre Markos University, 2021.

### Methods

Institution-based cross-sectional study design was employed from March 15–29, 2021. A simple random sampling technique was used to select 475 study participants. A binary logistic regression model was used to identify factors associated with mental distress. Variables with a p-value less than 0.25 in the bivariable analysis were entered into multivariable logistic regression analysis and a P-Value of less than 0.05 was considered as having a statistically significant association.

### Result

The magnitude of mental distress among students was found to be 35.4%, 95%CI (31%, 40%). Female sex [AOR = 1.95; 95%CI (1.24–3.06)], financial distress[AOR = 1.64; 95%CI

**Funding:** The authors received no specific funding for this work.

**Competing interests:** The author(s) declared no potential conflicts of interest with respect to the research, authorship, and/or publication of this article.

**Abbreviations:** AOR, Adjusted Odds Ratio; COR, Crud Odds Ratio; CGPA, Cumulative Grade point Average; DMU, Debre Markos University; SMCHS, School Of Medicine and College of Health Sciences; SPSS, Statistical Product Service and Solution; SRQ, Self-Reporting Questionnaire; USD, United State Dollars; WHO, World Health Organization.

(1.062.54)], feeling of insecurity [AOR = 2.49; 95% CI (1.13–3.54)], lack of interest to department [AOR = 2.00; 95%CI (1.75–4.36)] and cumulative grade point average less than expected [AOR = 2.63; 95%CI (1.59–4.37)]were significant variables with mental distress.

## Conclusion

The magnitude of mental distress was high. Sex, financial distress, feeling of insecurity, lack of interest in the department, and cumulative grade point average less than expected were significant variables with mental distress, so special attention on mental health promotion is required from policymakers, college officials, parents, and other Non-Governmental organizations.

## Introduction

Mental distress is a collection of mental health abnormalities that may not be group into standard diagnostic criteria and characterized by symptoms of anxiety, depression, insomnia, fatigue, irritability, forgetfulness, difficulty in concentrating, and somatic symptoms [1]. Depression, anxiety and stress are the most common forms of mental distress among university students [2]. Stress has been defined as a process in which environmental demands exceed the perceived capability of an individual to cope. Anxiety is characterized by cognitive, physical, emotional, and behavioral components. depression also defined as persistent sad, anxious, feelings of hopelessness, and feelings of guilt [3].

Mental distress is a public health issue explained with variable levels of stress, confused emotions, hallucination, depression, anxiety, panic or somatic symptoms. These symptoms may experience in persons without actually being ill in a medical sense and dominantly interfere students relation with their friends, life events and academic performance [4].

University students are special groups of the communities that are enduring a critical transitory period in which they are going from adolescence to adulthood, from high school to campus and can be one of the most stressful times in their life [5]. Also they represent a unique population with special concerns, obligations and worries that differ from other age and occupational groups [6, 7]. The gap between the need for treatment of mental disorders and its provision is wide all over the world. Many students in colleges and universities today have many different kinds of mental disorders for which they may, or may not, be seeking treatment [8].

The national alliance on mental Illness reports showed that one in five university students encounter mental distress, with 75% of all mental health conditions encounter by the age of 24 [9]. Ronald Kessler from Harvard University found that 37% of people aged 15–24 years have confirmed mental problems. Most of the University students today fit within this age group [10].

In Africa, mental distress is an important public health concern that is considered as a public health problem. Study conducted in South Africa revealed that the prevalence of common mental disorders is 27% [11].

In Ethiopia mental health abnormalities account for 11% of total burden of the disease conditions [12]. Studies examining the magnitude of mental distress in Ethiopia using different cutoffs points of the Self-reporting questionnaire (SRQ-20) have reported magnitude of mental distress ranges from 21.6 to 49.1% among university students [13, 14]. From the total burden of disease mental disorders accounts 25.3% and 33.5% of all years lived with a disability in low- and middle-income countries, respectively [15].

A number of factors including, academic responsibilities, separation from their family and environment, competition with their peers, workload, substance use, exposure to patients suffering and deaths could be reasons why mental distress is very common in students than the general population [8, 16, 17].

Mental distress has overwhelmed negative effect on role functioning like that of many serious chronic conditions [18]. Mental health conditions frequently lead individuals and families into poverty and decrease economic development at the country level [15]. Students with mental distress are at an increased risk of poorer general health [19].

In addition, there is a significant inverse relation between life satisfaction and mental distress among university students [20]. High prevalence, and risk factors of mental distress among clinical students not only affect their health but also affect their academic achievements [21].

Different studies showed that college students torched by their duties, encounter anxiety, depression, poor in academic performance and loosing academic responsibilities [22, 23]. Due to mental distress 55.8% Ethiopian University students had poor sleep quality [24].

The findings of this study will aid in the development of evidence-based mental health promotion programs for students and will be used as a resource to seek remedial action from policymakers, school of medicine and college of health science (SMCHS), non-governmental organizations, and other interested parties, thereby improving students' academic performance and patient care.

Despite its magnitude, severity and negative consequence, studies on magnitude of mental distress and associated factors among health science students in Ethiopia were limited, particularly in the study area, Debre Markos University (DMU).

## Methods

### Study design

An institution-based cross-sectional study design was used.

### Study area and study period

The study was conducted at DMU, Northwest Ethiopia. DMU is a public university located in the town of Debre Markos, which is located in Northern part of Ethiopia in Amhara region. It is 299 km far from the capital city of Ethiopia, Addis Ababa and 265 km far from the capital city of Amhara Region Bahir Dar. The University has seven Colleges with a total of 11,278 regular students. Under SMCHS there are 10 departments with 1537 (1019 male and 518 female) students [25]. The study was conducted from March 15–29, 2021.

### Source populations

The Source populations of the study were all regular undergraduate SMCHS students at DMU.

### Study population

The study population consists of all regular undergraduate SMCHS students who were available during the data collection time.

### Inclusion criteria

All regular undergraduate students at SMCHS were included in this study.

## Exclusion criteria

Critically ill students, students on summer break, and those with a history of mental illness were excluded from the study.

## Sample size

The sample size was calculated for both the first (to determine the magnitude of mental distress among undergraduate SMCHS students) and second specific objectives (to identify factors associated with mental distress among undergraduate SMCHS students). Using single population proportion formula with 95% level of confidence, 5% of marginal error, and taking prevalence of mental distress 40.9% which was done in University of Gondar [26] and including10% non-response rate, the sample size was 408.

$$n = \left( \frac{z_2^{\alpha^2} \times p(1-p)}{d^2} \right)$$

Where

**n**- Minimum sample size

**P**-Estimated proportion of evidence based practice (40.9%)

**d**-the margin of sampling error tolerated (5%)

**Zα**/2- is the standard normal distribution at 1-α% confidence level (95% = 1.96)

For the second objective, the sample size was determined using double population proportion formula by considering the following assumptions; power of 80%, 95% Confidence level and 5% level of precision and low grade than expected as a factor variable based on the study done in University of Gondar [26] and including10% non-response rate, the sample size was 475. So, this was selected for final sample size. The calculation was done by using Epi info version 7 statistical package.

## Sampling technique and procedure

Simple random sampling technique was used to approach the study participants. The students were stratified into ten strata based on their respective department. Then, the total sample size was calculated for each stratum based on probability of proportion population size in each department. Again the respective sample size from each study year of the department was calculated using proportionate allocation formula. Finally, the samples were selected from each study year of the department using simple random sampling technique (Fig 1).

## Variables

**Dependent variable.**   Mental distress.

**Independent variables.**   Age, sex, take part in religious practice, having boy or girl friend, conflict with boy or girl friends, family place of residence, educational status of the parents, marital status, number of family members, students monthly income, feeling of insecurity about safety, having financial support, family history of mental illness, having financial distress, exposure to violence, department, department choice, interest to department, vacation time, decrease grade than expected, conflict with instructors, year of study, social support and substance use were independent variables.

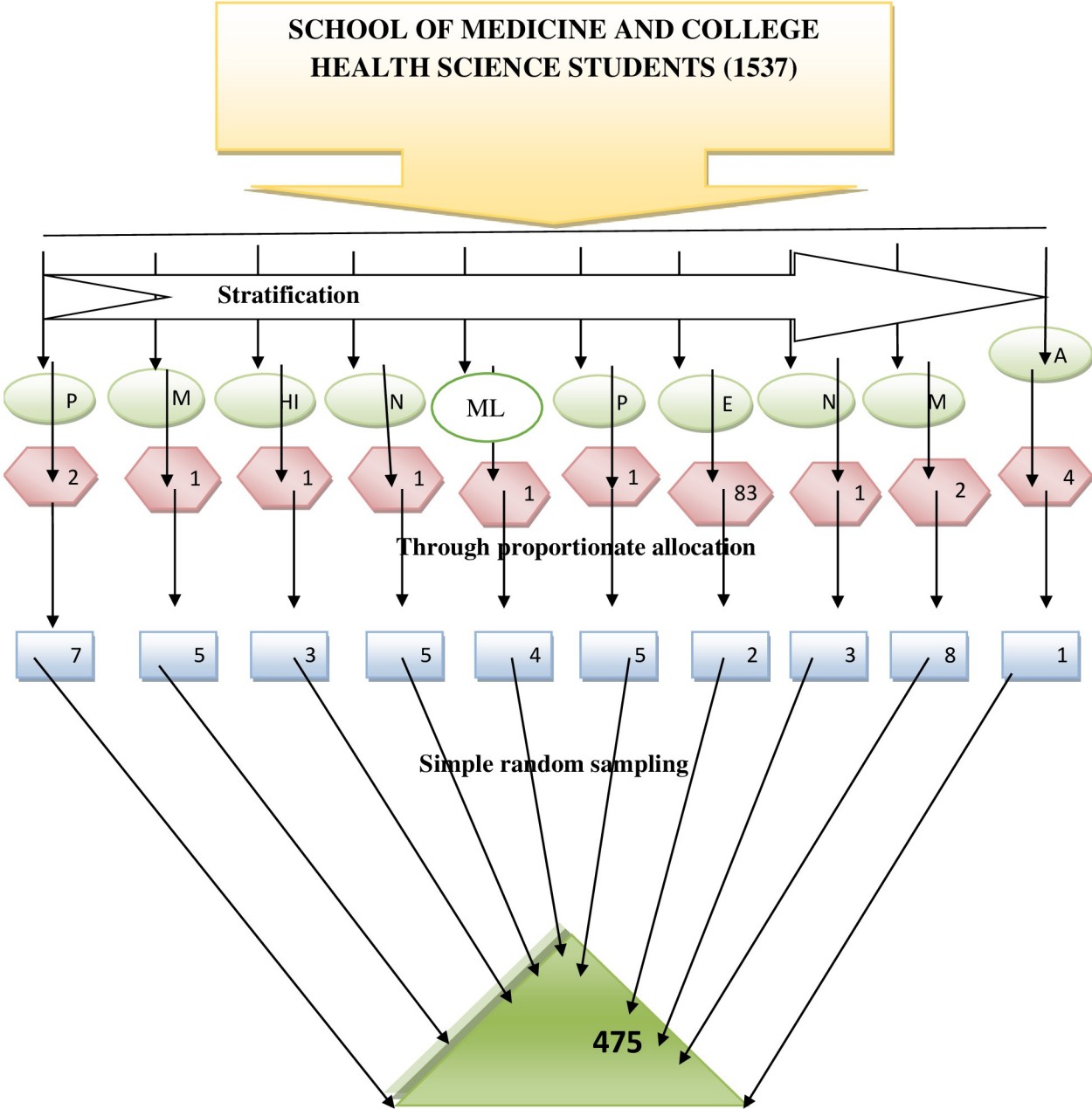

**Fig 1. Schematic presentation of sampling technique for mental distress and associated factors among SMCHS students at DMU, Northwest Ethiopia, 2021.** P = Pharmacy, M = Medicine, HI = health informatics, N = nursing, ML = Medical laboratory, P = Public health, E = enviromental health, N = nutrition, M = Midwifery, A = Anesthesia.

## Operational definition

**Mental distress**: Students who were found to have 8 or more symptoms out of the 20 SRQ items in the last 4 weeks were considered as having mental distress [14, 26–28].

**Current substance users**: when students use specified substance like khat, cigarette, cocaine, alcohol, hashish (for non-medical purposes) in the last one month [29].

**Low social support**- participants having <3 score in social support scoring scale, **Moderate social support**- participants having 3–5 score in social support scoring scale, and **Strong social support**- participants having >5 score in social support scoring scale [30].

**Feeling of insecurity**- sense of inadequacy and uncertainty in any aspect of academic activities

**Financial distress**- lack of money for all educational activities

## Data collection tools and procedures

Self-Reporting Questionnaire-20 (SRQ-20) is a standardized questionnaire having 20 items was used to indicate magnitude of mental distress. This tool developed by WHO and is validated in Ethiopia and other low socioeconomic countries. Data were collected by four data collectors and two supervisors after getting consent from the study participants. First, the questionnaire was prepared in English language and it was translated to Amharic and back to English by language experts for consistency and to easy of understanding. Amharic version of the questionnaire was used for data collection. Data were collected using structured self-administered questionnaire having five parts. The first part contains socio-demographic characteristics of students. The second part of the questionnaire is a SRQ-20 item to determine the magnitude of mental distress. The rest 3 parts of the questionnaire are questions related to Academics, substance use and social support.

## Data quality assurance and control

Training and orientation for the supervisors and data collectors were given. In order to evaluate the clarity of the questions in the questionnaire and to ensure that the reaction of the respondents to the questions, pretest was done on 5% of study subjects at Tropical College of Medicine before one week of the actual data collection period and internal consistency of the measuring tool was checked by Cronbachs alpha (CA = 0.86). Also appropriate modification on the questionnaire was done. The collected data were reviewed and checked for its completeness before data entry and incomplete data were discarded. Epi Data was used for data entry to prevent data entry errors.

## Data processing and analysis

The data were checked for its completeness and consistency. Then it were coded and entered in EPI data version 3.02 Software. After that, data were exported to SPSS version 25 for analysis. Model fitness test (Goodness of fit was checked with Hosmer Lemshow model of fit (p = 0.361) and assumption test for multicollinearity problem (VIF≤ 3.35) was done). Descriptive analysis using frequencies, proportions, graphs was performed to describe number and percentage of socio-demographic characteristics. Binary logistic regression analysis model was used to identify associated factors of mental distress. Variables with p-value less than 0.25 in the bivariable logistic regression were entered into the final multivariable logistic regression analysis to control possible confounding and in order to not miss biologically important variables. A p-value of less than 0.05 in multivariable regression analysis was considered as statistical significant with mental distress. The results were presented in text, tables and graphs based on the types of data.

## Ethical considerations

Ethical clearance was obtained from Debre Markos University Health Science College Ethical Review Committee. Written consent was used. Also affirmation that they are free to withdraw

consent and to discontinue participation was made. Privacy and confidentiality of collected information was ensured throughout the process as no name is written. Privacy and confidentiality of collected information was ensured throughout the process as no name is written. This study was conducted in accordance with the declaration of Helsinki.

## Result

### Socio-demographic characteristics

Among 475 study participants, 461 students were participated in the study giving a response rate of 97.1%; of these 295(64%) were males. Nearly two-thirds of students (68.5%) were between the ages of 20 and 24, with the median age of the respondents being (21–25 IQR) years. Most of the respondents were Orthodox religion followers (92.6%), 14(3.1%) Muslim and the rest were protestant, 93.9% were from Amhara ethnic group and 88.5% were single. Majority (88.5%) of the participants were married and the rest were single. Among the participants, 88.9% had no family history of mental illness, 50.3% had no financial distress, 84.8% had feeling of insecurity, 93.8% had no exposure to violence, 93.5% had financial support, 23.9% had no conflict with boy or girl friend, 49.7 had financial distress, and 64.6% had no boy or girl friend (Table 1).

### Academic characteristics of students

One hundred thirty one (28.4%) of the respondents were first year and 124 (26.9%) were second year. Among the participants 77(16.7%) were from medicine and 66(14.3%) were from public health department (Fig 2). Almost three forth of the students 345 (74.8%) have entered their department of choice, and 336 (72.9), have expressed interest in their department (Table 2).

**Table 1. Socio demographic characteristics of DMU SMCHS students on mental distress, 2021 (N = 461).**

| Variables | Category | Frequency(n) | Percent (%) |
|---|---|---|---|
| Take part in religious practice | Always | 136 | 29.5 |
| | Usually | 112 | 24.3 |
| | Sometimes | 191 | 41.4 |
| | Never | 22 | 4.8 |
| Educational status of the father | Can't read and write | 142 | 30.8 |
| | Can read and write | 204 | 44.3 |
| | Primary | 40 | 8.7 |
| | Secondary | 34 | 7.4 |
| | College and above | 41 | 8.9 |
| Educational status of the mother | Can't read and write | 258 | 56.0 |
| | Can read and write | 120 | 26.0 |
| | Primary | 22 | 4.8 |
| | Secondary | 30 | 6.5 |
| | College and above | 31 | 6.7 |
| Number of family members | <4 | 25 | 5.4 |
| | 4–7 | 301 | 65.3 |
| | >7 | 135 | 29.3 |
| Monthly income of the respondent | ≤400 | 193 | 41.9 |
| | 401–700 | 123 | 26.7 |
| | ≥701 | 145 | 31.5 |

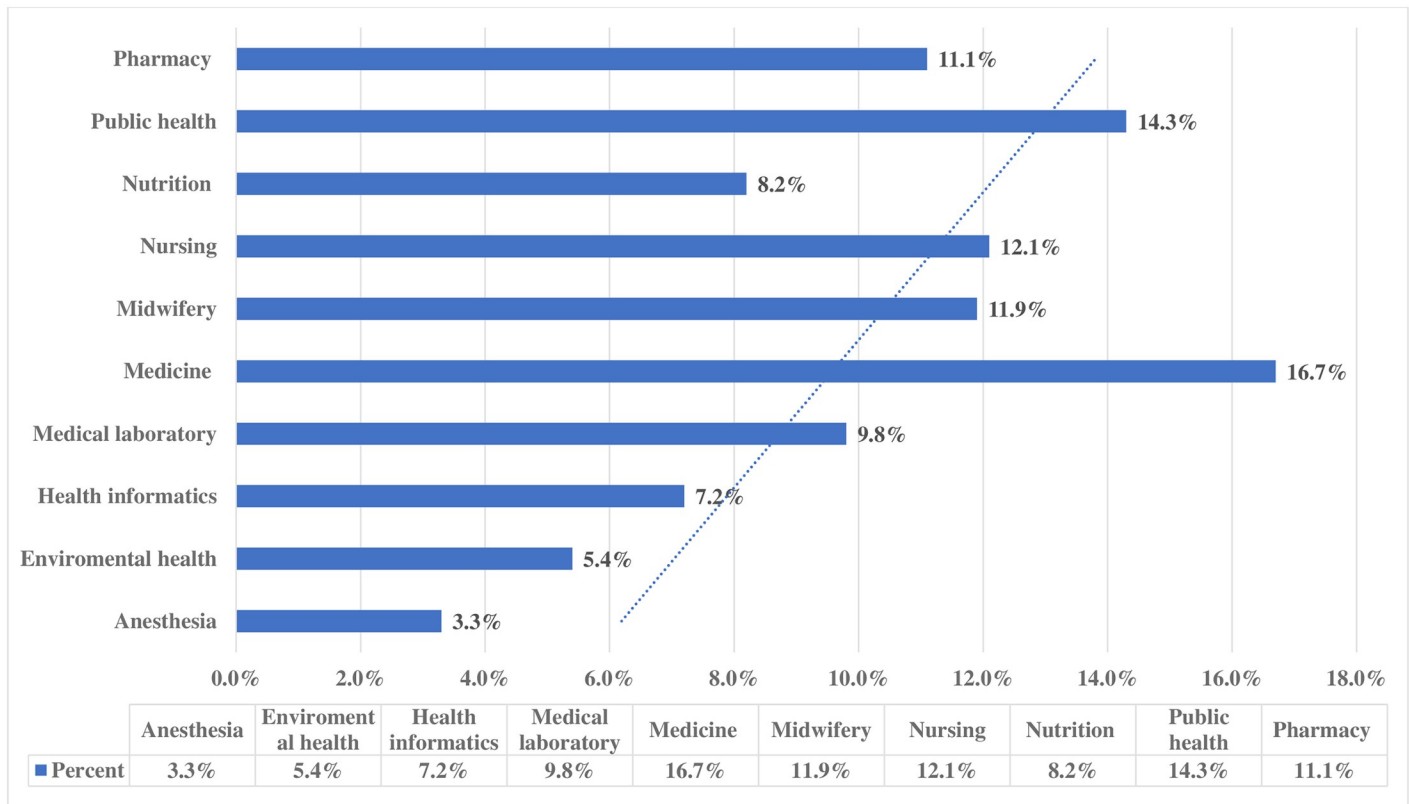

**Fig 2. Showing department among DMU SMCHS students on mental distress North West, Ethiopia, 2021 (N = 461).**

## Levels of social support

A level of social support is computed from the means of 12 items of multi-dimensional social support questions. Mean of special support, family support, friend support and over all social support become 3.2, 5.55, 4.6, and 4.4 respectively (Fig 3).

**Table 2. Academic related characteristics of DMU SMCHS students on mental distress, North West, Ethiopia, 2021(N = 461).**

| Variables | Category | Frequency(n) | Percent(%) |
|---|---|---|---|
| Year of study | Year one | 131 | 28.4 |
| | Year two | 124 | 26.9 |
| | Year three | 99 | 21.5 |
| | Year four | 81 | 17.6 |
| | Year five | 26 | 5.6 |
| Department choice | Preferred | 345 | 74.8 |
| | not preferred | 116 | 25.2 |
| Interest to the department | Yes | 336 | 72.9 |
| | No | 125 | 27.1 |
| Conflict with instructors | No | 407 | 88.3 |
| | Yes | 54 | 11.7 |
| Enough vacation time | No | 233 | 50.5 |
| | Yes | 228 | 49.5 |
| CGPA decrease than expected | Yes | 323 | 70.1 |
| | No | 138 | 29.9 |

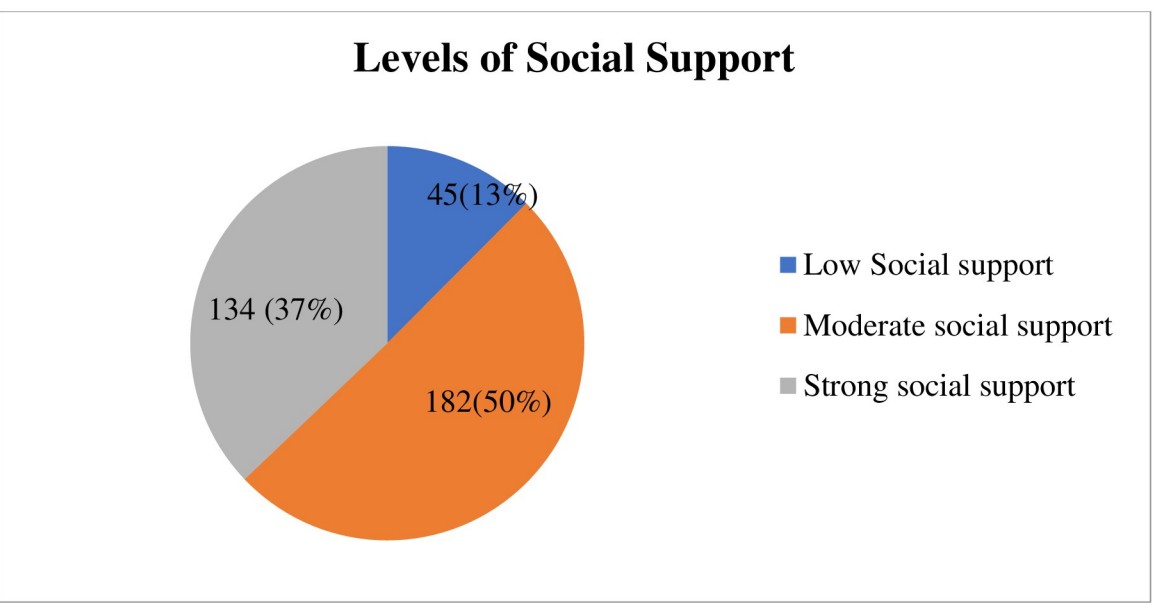

**Fig 3. Showing social support level among DMU SMCHS students on mental distress North West, Ethiopia, 2021 (N = 461).**

### Substance use related factors

Among respondents 125(27.11%) of them use alcohol (Fig 4).

### Magnitude of mental distress

Over all 35.4%, 95% CI, (31–40%) of students had mental distress (Fig 5).

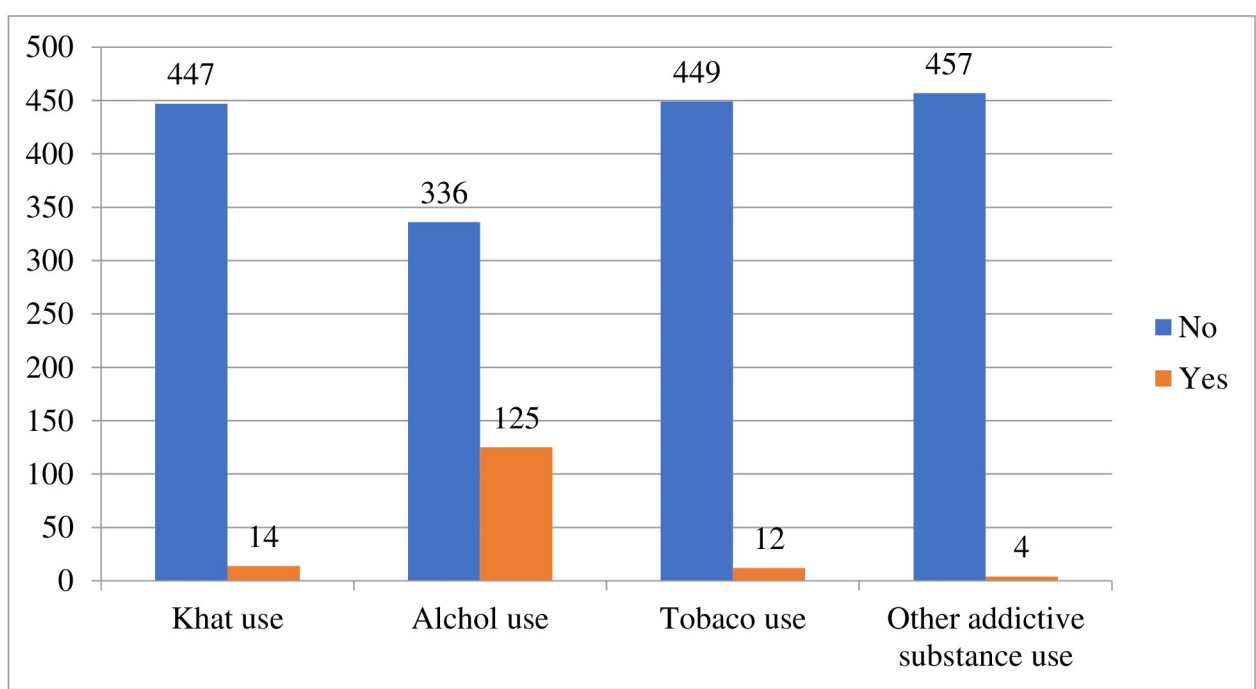

**Fig 4. Substance use among DMU school SMCHS students on mental distress, North West, Ethiopia, 2021 (N = 461).**

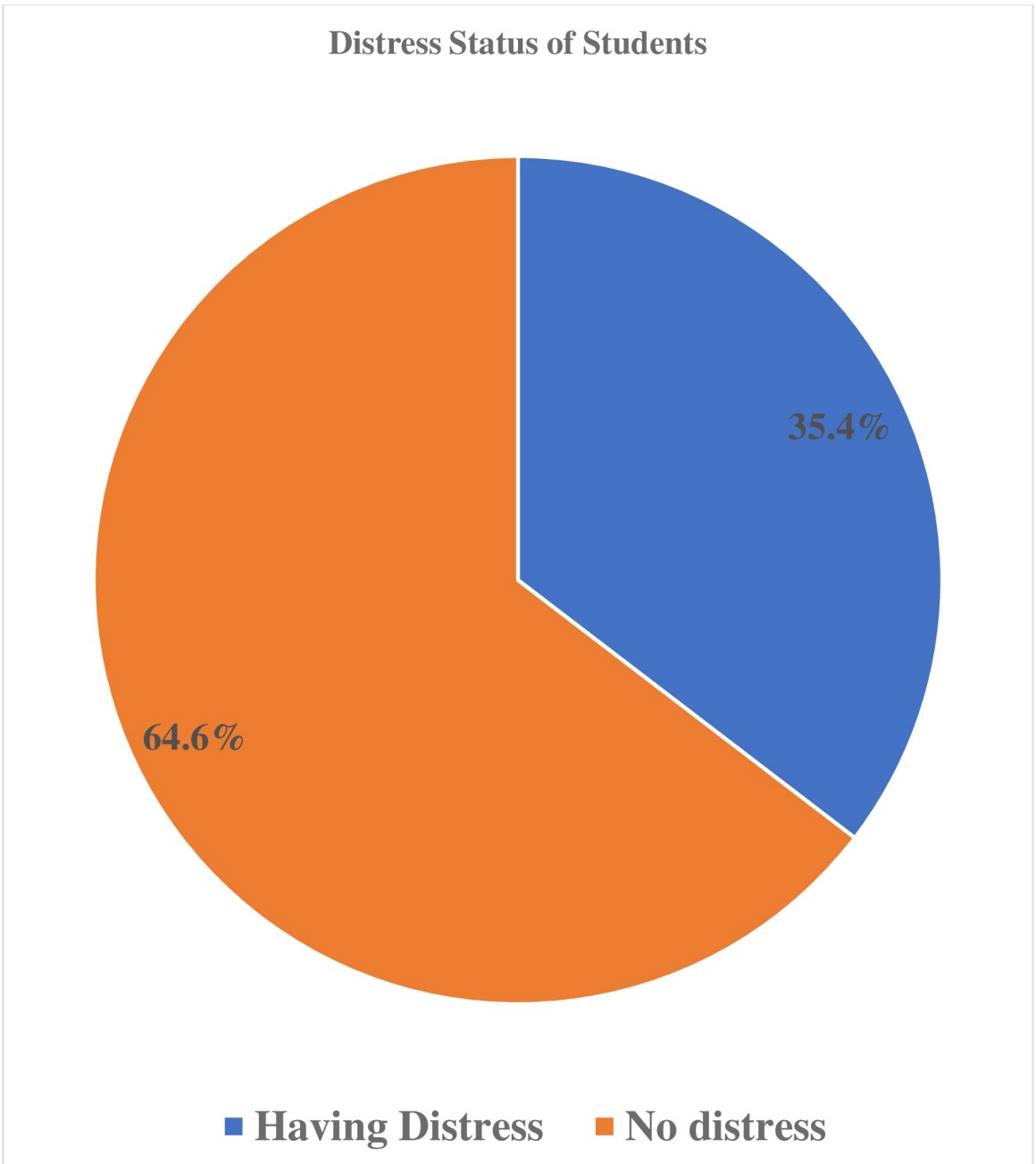

**Fig 5. Showing magnitude of mental distress among DMU school SMCHS students on mental distress, North West, Ethiopia, 2021 (N = 461).**

### Factors associated with mental distress

In multivariable logistic regression analysis, covariates like sex, financial distress, feeling of insecurity, lack of interest to the department and cumulative grade point average less than expected were statistically significant factors of mental distress among DMU School of medicine and college of health science students.

**Table 3. Bivariable and multivariable logistic regression analysis of factors associated with mental distress those statically associated among DMU SMCHS students on mental distress, North West, Ethiopia, 2021(N = 461).**

| Variable | | Mental distress | | COR 95% CI | AOR 95% CI | P value |
|---|---|---|---|---|---|---|
| | | Yes | No | | | |
| Sex | Female | 67 | 99 | 1.40 (0.94–2.08) | 1.95(1.24–3.06) | 0.04 |
| | Male | 96 | 199 | 1 | 1 | |
| Financial distress | Yes | 95 | 134 | 1.71(1.16–2.51 | 1.64(1.06–2.54) | 0.03 |
| | No | 68 | 164 | 1 | 1 | |
| Feeling of insecurity | Not secured | 37 | 33 | 2 (1.41–3.95) | 2.49(1.13–3.54) | 0.02 |
| | Secured | 126 | 265 | 1 | 1 | |
| | No | 146 | 285 | 1 | 1 | |
| | Preferred | 106 | 239 | 1 | 1 | |
| Interest to the Department | No | 69 | 56 | 3.17 (2.07–4.85) | 2.00 (1.75–4.36) | 0.001 |
| | Yes | 94 | 242 | 1 | 1 | |
| | No | 129 | 278 | 1 | 1 | |
| Decrease CGPA than Expected | Yes | 136 | 187 | 2.99 (1.86–2.12) | 2.63 (1.59–4.37) | 0.001 |
| | No | 27 | 111 | 1 | 1 | |
| | Moderate | 102 | 180 | 1.54 (0.98–2.43) | 1.36 (0.52–3.54) | |
| | Strong | 36 | 98 | 1 | | |

Being female was about 2 times [AOR = 1.95; 95%CI (1.24–3.06)] more likely to develop mental distress. Students having financial distress were 1.64 times [AOR = 1.64; 95%CI (1.06–2.54)], more likely to develop mental distress as compared to those who didn't have financial distress. Students who had feeling of insecurity were about 2.5 times [AOR = 2.49; 95% CI (1.13–3.54) more likely to develop mental distress than their counter parts.

Those students who didn't have interest to the department were 2 times [AOR = 2.00; 95% CI (1.75–4.36)] more likely to develop mental distress as compared to those having interest to their department. Students whose cumulative grade point average less than expected were about 2.6 times [AOR = 2.63; 95%CI (1.59–4.37)]more likely to develop mental distress as compared with their counter parts (Table 3).

## Discussion

The aim of this study is to assess the magnitude and associated factors of mental distress among DMU School of Medicine and College of Health sciences students. A high prevalence of mental distress among students is a cause of concern as it may alter behavior of students, diminish performance, and ultimately affect patient care and social service after their graduation.

**The theoretical values of this finding**- it expand body of knowledge for interested readers regarding on mental distress, identifying the possible risk factors and determining the magnitude of mental distress among students.

**Practical values of this finding**-the university may prepare an action plan for promoting mental health through counseling service, professionals give due attention and assess mental health problems when they assess physical and physiological complains, it is also important for meta-analysis and used as an input for policy making and researchers use this finding as a baseline for future study.

In this study 35.4% (95%CI, 31–40%) of students had mental distress and significantly associated with sex, financial distress, feeling of insecurity, lack of interest to the department, and cumulative grade point average less than expected.

In this study the prevalence of mental distress was 35.4%. This finding was consistent with study conducted in Saudi Arabia (Majmaah) university,35.8% [31] and Saudi Arabia, Jazan, university,31% [32]. This similarity may be due to use of the same measuring tool (SRQ-20).

Studies examining the prevalence of mental distress in Ethiopia using varying cuts off the Self-reporting questionnaire (SRQ-20) have reported prevalence of mental distress ranges from 21.6 to 49.1% among university students [13, 14]. This study result also came up with this range.

This consistency may be due to use of the same measuring tool and similar study subjects in similar setting, Ethiopia.

This study finding was lower as compared with study done at Gondar university 40.9% [26]. This might be due to time variation, difference in study subject, sample size, the improvement of infrastructure and a service option provided by the university from time to time.

Also this study finding was lower as compared with studies conducted in Australia,65.2% [33], Arish,41.9% [34], Nigeria,47.3% [35], and India, Malaysia 48.3% [36] university students. This difference may be contributed to variation in the curriculum, measuring tool, living condition and other constraints.

This study was higher than studies conducted in Mizan Aman health science college students (29.2%) [29] and Hawassa university medical students (30%) [14], Adama University students(21.6%) [37]. This difference might be attributed to slight difference in study subjects (only medical students in Hawassa University, all university students in Adama and college students in Mizan Aman), use of SRQ-20 cut point(11 was used in Mizan Aman Health Science College and Adama) and infrastructures of the institution, time of study (2013 in Adama, before 2016 at Hawassa). Further, this study was done among health science students where the health science education environment is more of stressful and contributes to mental distress, but in Adama it was conducted among all University students.

Also the current study finding was higher as compared with study conducted in Punjab, India (15%) [38] Somaliland, Hargeisa (19.8%) [39] and Egypt, Assiut (17%) [28] university students. This variation may be due to variation in study time (which was at 2015 in India), socioeconomic condition, life style and environmental factors, variation in study subjects (only medical students were studied in India and Hargesia and all college students in Assiut), use of SRQ-20 cutoff point for measurement (India uses 10 and Hargesia uses 11).

The likelihood of mental distress was about 2 times higher among female students (AOR = 1.95) than male students. This finding is in line with studies done in Malaysian [36], India [38], Majamaah [31], Saudi Arabia, Jazan [32] Assiut [28] Mizan Aman Health Science College [29]. This may be due to the fact that affective nature of females in response to stressors, domestic violence, and hormonal changes during menstruation.

Students having financial distress were 1.64 times (AOR = 1.64), more likely develop mental distress as compared to those who didn't have financial distress. This result is consistent with studies conducted in Mizan Aman health science college students (29.2%) [29] and Gondar [26] University students. This could be due to the rising cost of stationary materials and photocopy services may create stressful condition in students. Moreover, students with financial difficulty develop anxiety, frustration, and sense of hopelessness and difficulty of sleeping which may further lead students mentally distressed.

Students who had feeling of insecurity were about 2.5 times (AOR = 2.49) more likely develop mental distress than their counter parts. This study is in line with study conducted among Hawassa University medical students [14].

In this study, students who didn't have interest to their department were 2 times more likely to develop mental distress as compared with those who have interest with their department. This finding is in agreement with study done in Gondar University [26]. This might be due

poor achievement in academics since those students who have no interest to their field of study may not be initiated to read more and lost their time.

Students whose grades lower than expected were about five times more likely to develop mental distress than their counter parts. This study result was supported by other study done in Gondar University students [26]. This might be due to the fact that students whose grades lower than expected may experience anxiety, frustration, and sense of hopelessness and difficulty of sleeping which may further lead students mentally distressed.

## Limitations of the study

Recall bias may be there since most of the questions assess past history. The study may be prone to reporting bias since the data were collected based on self-reported information. This study investigated mental distress rather than specific mental health disorders. Therefore, mental distress in this study may represent those with mental disorders or those who experience a temporary distress due to situational stresses.

## Conclusion

Prevalence of mental distress was high. Sex, financial distress, felling of insecurity, lack of interest to the department and cumulative grade point average less than expected were statistically significant with mental distress.

Since the magnitude of mental distress is high, health professionals working in student's clinic should be aware that many students complaining of physical symptoms may also be suffering from mental health problems, and thus they better assess for both these issues.

It is also recommended that mental distress needs due attention and remedial action from policy makers, ministry of health, non-governmental organizations, and other concerned bodies to enhance students' performance on academic and ultimately proper patient care after their graduation. Further research with qualitatively study design should be conducted for further information using this finding as a secondary data.

## Supporting information

**S1 File.**
(DOCX)

**S1 Data.**
(SAV)

## Acknowledgments

We are grateful to the participants from Feresebet Town's food and beverage establishments who provided us with the essential information. We'd also want to thank the data collectors and supervisors for their time and effort.

## Author Contributions

**Conceptualization:** Baye Tsegaye Amlak, Mezinew Sintayehu Bitew, Asmamaw Getnet, Fentahun Minwuyelet Yitayew, Tamene Fetene Terefe, Tadesse Tsehay Tarekegn, Asmare Getie Mihret, Omega Tolessa Geleta, Fisha Alebel GebreEyesus, Dejen Tsegaye.

**Data curation:** Baye Tsegaye Amlak, Mezinew Sintayehu Bitew, Asmamaw Getnet, Fentahun Minwuyelet Yitayew, Tamene Fetene Terefe, Tadesse Tsehay Tarekegn, Omega Tolessa Geleta, Gebrie Getu Alemu, Fisha Alebel GebreEyesus.

**Formal analysis:** Baye Tsegaye Amlak, Mezinew Sintayehu Bitew, Asmamaw Getnet, Asmare Getie Mihret, Fisha Alebel GebreEyesus, Dejen Tsegaye.

**Funding acquisition:** Baye Tsegaye Amlak, Mezinew Sintayehu Bitew, Asmare Getie Mihret, Gebrie Getu Alemu.

**Investigation:** Mezinew Sintayehu Bitew, Asmamaw Getnet, Tamene Fetene Terefe, Asmare Getie Mihret, Omega Tolessa Geleta, Fisha Alebel GebreEyesus, Dejen Tsegaye.

**Methodology:** Baye Tsegaye Amlak, Mezinew Sintayehu Bitew, Asmamaw Getnet, Fentahun Minwuyelet Yitayew, Tamene Fetene Terefe, Tadesse Tsehay Tarekegn, Asmare Getie Mihret, Omega Tolessa Geleta, Gebrie Getu Alemu, Fisha Alebel GebreEyesus, Dejen Tsegaye.

**Project administration:** Mezinew Sintayehu Bitew, Asmamaw Getnet, Tadesse Tsehay Tarekegn, Asmare Getie Mihret.

**Resources:** Fentahun Minwuyelet Yitayew, Tamene Fetene Terefe, Asmare Getie Mihret, Omega Tolessa Geleta, Gebrie Getu Alemu.

**Software:** Baye Tsegaye Amlak, Fentahun Minwuyelet Yitayew, Tamene Fetene Terefe, Asmare Getie Mihret, Omega Tolessa Geleta.

**Supervision:** Asmamaw Getnet, Fentahun Minwuyelet Yitayew, Tadesse Tsehay Tarekegn, Asmare Getie Mihret, Gebrie Getu Alemu.

**Validation:** Asmamaw Getnet, Fentahun Minwuyelet Yitayew, Dejen Tsegaye.

**Visualization:** Fentahun Minwuyelet Yitayew, Tadesse Tsehay Tarekegn, Gebrie Getu Alemu.

**Writing – original draft:** Baye Tsegaye Amlak, Tadesse Tsehay Tarekegn, Omega Tolessa Geleta, Dejen Tsegaye.

**Writing – review & editing:** Baye Tsegaye Amlak, Mezinew Sintayehu Bitew, Asmare Getie Mihret, Omega Tolessa Geleta, Gebrie Getu Alemu, Dejen Tsegaye.

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
