## [Decision Letter · Decision Letter 0]

23 May 2022

PONE-D-22-05826THE MAGNITUDE OF MENTAL DISTRESS AND ASSOCIATED FACTORS AMONG A SCHOOL OF MEDICINE AND COLLEGE OF HEALTH SCIENCES STUDENTS AT DEBRE MARKOS UNIVERSITY, 2021PLOS ONE

Dear Dr. Tsegaye,

Thank you for submitting your manuscript to PLOS ONE. After careful consideration, we feel that it has merit but does not fully meet PLOS ONE’s publication criteria as it currently stands. Therefore, we invite you to submit a revised version of the manuscript that addresses the points raised during the review process.

We look forward to receiving your revised manuscript.

Kind regards,

Matias Noll, Ph.D

Academic Editor

PLOS ONE

Journal Requirements:

"First of all, we would like to thank Wolkite University for financial support.

Our heartfelt gratitude also goes to, Debre Markos University college of medicine and health science undergraduate students who participate in the study

We would also give our appreciation to data collectors and supervisors for their endeavor."

4. Thank you for stating the following in the Funding Section of your manuscript: 

"This study was funded by Wolkite University. The funders had no role in study design, data collection and analysis, decision to publish, or preparation of the manuscript"

We note that you have provided funding information that is not currently declared in your Funding Statement. However, funding information should not appear in the Funding section or other areas of your manuscript. We will only publish funding information present in the Funding Statement section of the online submission form. 

Reviewers' comments:

Reviewer's Responses to Questions

**Comments to the Author**

1. Is the manuscript technically sound, and do the data support the conclusions?

Reviewer #1: Partly

Reviewer #2: Yes

Reviewer #3: Partly

Reviewer #4: Yes

2. Has the statistical analysis been performed appropriately and rigorously? 

Reviewer #1: Yes

Reviewer #2: Yes

Reviewer #3: Yes

Reviewer #4: Yes

3. Have the authors made all data underlying the findings in their manuscript fully available?

Reviewer #1: Yes

Reviewer #2: No

Reviewer #3: Yes

Reviewer #4: Yes

4. Is the manuscript presented in an intelligible fashion and written in standard English?

Reviewer #1: No

Reviewer #2: No

Reviewer #3: No

Reviewer #4: Yes

5. Review Comments to the Author

Reviewer #1: Abstract:

• Introduction is too long; it should be short. The instrument used to assess mental distress in this study should be

mentioned.

• Key words: There were only three keywords, which may be increased.

Introdcution:

• At the end of the introduction, state the purpose of the study.

Methods:

• Line-144: "Study participants were not excluded from this study." This is not an exclusion criteria.

• Line-146: Earlier, no objectives were mentioned.

• Line 162-163: There is no need to mention the figure title if there is no figure at this point.

• Line 181: In the definition, what type of "specified substance" should be mentioned?

• Line 222: The reference/Id number of the ethical approval should be given.

Results:

• Line 238: There is no need to include 'North West Ethopia' in the Tables title.

• Table 1 shows no consistency in the variables and categories. They need to be rearranged (e.g against the age group

in the variable column, the religion of the students was mentoned in the category column.

• Table-1 is too long

• Line 243-244: ‘Nearly three fourth of the students ……. have interest to their department’ This sentence is

perplexing because of the words ‘with their choice ' and ‘have interest '.

• Line 257-258: There is no need to mention figure title without a figure.

• All the figures are mentioned as Figure No-1.

Discussion:

• Line 303-304: ‘similar study subjects in similar setting, Ethiopia.’ is unclear.

• Line 319-320: ‘health science education environment is more of stressful’ is unclear, need explanation.

• There are some more explanations that are not supported by other studies or the findings of this study.

Conclusion:

• The last paragraph (372–375), not supported by the findings of this study, should be removed from conclusion

References:

• Many of the references are not properly written, and page numbers are missing.

Opinion

• Overall, the manuscript requires improvements in the English language, including grammar.

• The manuscript requires major revisions.

Reviewer #2: 1. What are the exclusion criteria of the study? It has to be written clearly

2. There was merging of variables in table 1 see the rows at Educational status of the mother

3. Spacing problems for instance table 1 and the next paragraph and below table 2

4. Inappropriate or unimportant part in table 3 column 2 part, please check and correct

5. The letters and numbers in figures1 has to be given a key

6. The fonts in the figures must be similar with the main text

Reviewer #3: Authors have work to improve the quality of this study. The discussion part require some improvement. it follows the same fashion just by comparing the findings and then saying lower and higher than this and that. more over some the conclusions and recommendations are not based on the result example; proper time management and the one recommended for the University clinic workers. Overall the current form of manuscript can be suitable for this reputed journal i.e., Plos One with some improvement.

Reviewer #4: The manuscript reported the magnitude of mental distress and associated factors among a school of medicine and college of health sciences students at a university, which is an important topic

The manuscript was clear, generally well-written, structures and organized.

The Introduction provided a comprehensive background. However, grammatical and spelling mistakes presents. I advise to review the whole manuscripts for this particular point.

The methods section was specific. Details that supports the reproducibility of the research is presented very-well. The following was a concern:

-Why only undergraduate students were included? To me, post graduate might exhibit distress as well and might be sometime more than undergraduates as they could have families, work or other commitments. Any explanation?

- I don’t see there is a need for figure 1

The results section is thorough, well-subdivided and appropriately supported by data analyses.

- Please remove Muslim and protestant from age category and mention them as standalone variable- religion.

- I cannot read under “Educational status of the mother” in page 12; I believe there is missing variables which correspond to the mentioned categories, please review.

- Please re-order figures, all are 1? As a reader couldn’t correlate them easily.

- NO need for level of social support; can be written in text only.

- The last figure can be easily demonstrated in a pie chart

- Proper formatting is needed.

The Discussion section is well-sourced and comprehensive. However, future implications and research are missed.

The Conclusion provide a clear summation and identify the next steps for future research. However, it is lengthy little bit. Try to be more concise

6. PLOS authors have the option to publish the peer review history of their article (what does this mean?). If published, this will include your full peer review and any attached files.

Reviewer #1: **Yes: **Sk Akhtar Ahmad

Reviewer #2: No

Reviewer #3: No

Reviewer #4: No

---

## [Author Response · Author response to Decision Letter 0]

28 Jun 2022

Author’s Point-by-Point Response to the Reviewer's and Editors Reports

The magnitude of mental distress and associated factors among A school of medicine and college of health sciences students at Debre Markos university, 2021

Corresponding Authors Dejene Tsegaye/ dejenetsegaye@gmail.com

Point by point response to Reviewers and Editors 

First and foremost, the authors would like to express their gratitude to the PLOSE ONE Journal editors and reviewers for thoroughly evaluating this work and offering the required corrections. We made changes based on the feedback we received and presented each comment point by point. The authors attempted to address all of the concerns expressed by the editorial board and reviewers. Please note that the response was written in blue font.

Authors' responses to the editors' remarks

Response to the financial issue: This work was not funded in any way by the author(s). We also included it in the main document and in the cover letter.

Response to the Acknowledgement section: Wolkite University was mentioned as a source of financial assistance. However, this did not imply that Wolkite University had supported our research; it was mentioned in error. The necessary changes to the main document have been made, and our cover letter has been attached.

Response to the data availability issue: The data used to summarize this work are in the possession of the relevant author, and anybody can obtain them with a fair request.

Authors' responses to the authors' remarks

REVIEWER #1

ABSTRACT: 

Comment: Introduction is too long; it should be short. The instrument used to assess mental distress in this study should be mentioned.

Response: Thank you for your advice. We made the necessary changes.

Comment: Key words: There were only three keywords, which may be increased.

Response: Thank you for your suggestions. We have added one additional keyword that is crucial.

Comment: Introduction: At the end of the introduction, state the purpose of the study.

Response: It is critical to include the study's goal. Thank you for your insightful suggestion. We've added a paragraph that explains it.

METHODS:

Comment: Line-144: "Study participants were not excluded from this study." This is not an exclusion criteria.

Response: We didn't have any criteria in place to reject students from the research because of this. Critically ill students, students on summer break, and those with a history of mental illness were all eliminated from the study as a result of your suggestion.

Comment: Line-146: Earlier, no objectives were mentioned.

Response: We did not include objectives in the manuscript because we followed the journal's guidelines. We've put the objectives in brackets now, as per your advice. Thank you for taking the time to leave such an insightful comment.

Comment: Line 162-163: There is no need to mention the figure title if there is no figure at this point.

Response: Thank you for your input; it was inadvertently mentioned there. We've made a change as a result of your idea.

Comment: Line 181: In the definition, what type of "specified substance" should be mentioned?

Response: We indicated substances that a student may have taken in the previous month under designated substances. Thank you for providing us with such a useful comment.

Comment: Line 222: The reference/Id number of the ethical approval should be given.

Response: Under ethics approval and consent to participants, the ethical approval reference number has been mentioned, thank you. It was HSC/685/16/19.

RESULTS:

Comment: Line 238: There is no need to include 'North West Ethiopia' in the Tables title.

Response: Thank you for your advice. 'North West Ethiopia' has been omitted from all table tittles.

Comment: Table 1 shows no consistency in the variables and categories. They need to be rearranged (e.g against the age group in the variable column, the religion of the students was mentioned in the category column.

Response: Sure, the table needs to be rearranged, and the appropriate adjustments have been made.

Comment:Table-1 is too long

Response: Thank you for your kind feedback. The table was indeed excessively long, and variables that were presented in the description session have been deleted from the table, as per your recommendation.

Comment: Line 243-244: ‘Nearly three fourth of the students ……. have interest to their department’ This sentence is perplexing because of the words ‘with their choice ' and ‘have interest '.

Response: The remark implied that students in various departments were studying their departments despite their lack of enthusiasm. In any case, we made grammatical corrections.

Comment: Line 257-258: There is no need to mention figure title without a figure.

Response: Thank you for your input; it was inadvertently mentioned there. We've made a change as a result of your idea.

Comment: All the figures are mentioned as Figure No-1.

Response: On each figure's reference, the appropriate modification is made.

CONCLUSION:

Comment: The last paragraph (372–375), not supported by the findings of this study, should be removed from conclusion

Response: Based on your insightful comment, the paragraph concerning recommendations (the last paragraph of conclusion) has been delated.

REFERENCES:

Comment: Many of the references are not properly written, and page numbers are missing.

Response: Thank you for taking the time to leave such an insightful comment. Following our attempt to look over the references, we made the necessary revisions.

REVIEWER #2

Comment 1. What are the exclusion criteria of the study? It has to be written clearly

Response: We updated exclusion criteria based on your and another reviewer's comments, which include students on summer break and those with a history of mental illness. These criteria were not included in the exclusion criteria section since they were already factored into the non-participant rate.

Comment 2. There was merging of variables in table 1 see the rows at Educational status of the mother

Response: After going over the original paper, we made the necessary changes. When the manuscript was being produced, the correspondent author made a mistake, sorry.

Comment 3. Spacing problems for instance table 1 and the next paragraph and below table 2

Response: We made equivalent space across the document after all of the comments were fixed.

Comment 4. Inappropriate or unimportant part in table 3 column 2 part, please check and correct

Response: Yes, the variables in Table 3, column 2 and row 2 were inaccurate, and we fixed them. Thank you very much.

Comment 5. The letters and numbers in figures1 has to be given a key

Response: We appreciate your thorough inquiry; we corrected them based on your and other reviewers' comments.

Comment 6. The fonts in the figures must be similar with the main text

Response: It is a valuable comment, and we created it based on it.

REVIEWER #3

Authors have work to improve the quality of this study. The discussion part require some improvement. it follows the same fashion just by comparing the findings and then saying lower and higher than this and that. more over some the conclusions and recommendations are not based on the result example; proper time management and the one recommended for the University clinic workers. Overall the current form of manuscript can be suitable for this reputed journal i.e., Plos One with some improvement.

Response: We appreciate your suggestion. We did a number of things to improve the quality of the study after reevaluating the original document, starting with English language correction. We also attempted to update the discussion section in order to improve the discussion's flow. Inappropriate/unimportant thoughts are omitted from the conclusion and recommendation sections, since we had a similar request from another reviewer. Thank you so much for your insightful comment, which will assist us in improving the overall document quality.

REVIEWER #4

The manuscript reported the magnitude of mental distress and associated factors among a school of medicine and college of health sciences students at a university, which is an important topic

The manuscript was clear, generally well-written, structures and organized. 

The Introduction provided a comprehensive background. However, grammatical and spelling mistakes presents. I advise to review the whole manuscripts for this particular point.

The methods section was specific. Details that supports the reproducibility of the research is presented very-well. The following was a concern:

Comment: Why only undergraduate students were included? To me, post graduate might exhibit distress as well and might be sometime more than undergraduates as they could have families, work or other commitments. Any explanation?

Response: We agree with you; actually, post-graduate students may be distressed. However, university for postgraduate students is a second exposure, and they may not have the same mental suffering as undergraduate students in terms of the surroundings, finances, interest in their area, feelings of insecurity, and other factors.

Comment: I don’t see there is a need for figure 1

Response: We removed it, thank you.

The results section is thorough, well-subdivided and appropriately supported by data analyses.

Thank you for your helpful feedback.

Comment: Please remove Muslim and protestant from age category and mention them as standalone variable- religion.

Response: It was a blunder made during the preparation of the manuscript. We received similar feedback from other reviewers as well, and implemented the necessary changes after reevaluating the original document.

Comment: I cannot read under “Educational status of the mother” in page 12; I believe there is missing variables which correspond to the mentioned categories, please review.

Response: Certain factors were merged with the educational status variable in table 1. Now that the changes have been made, it is well written.

Comment: Please re-order figures, all are 1? As a reader couldn’t correlate them easily.

Response: Thank you very much; that was a complete blunder that has now been rectified.

Comment: NO need for level of social support; can be written in text only.

Response: We received a helpful suggestion from a reviewer to shorten the table. Many variables from the tables are now written in text form, as you suggested. Thank you so much for everything.

Comment: The last figure can be easily demonstrated in a pie chart

Response: As per your recommendation, we accomplished it in Pichart.

Comment: Proper formatting is needed.

Response: After a thorough examination, suitable formatting is applied to the entire manuscript.

Comment: The Discussion section is well-sourced and comprehensive. However, future implications and research are missed.

Response: Thank you for your encouraging words. Future implications have been added, which were actually discussed in the recommendation section.

Comment: The Conclusion provide a clear summation and identify the next steps for future research. However, it is lengthy little bit. Try to be more concise

Response: Thank you for your kind words. We tried to condense the concluding section by avoiding irrelevant information.

We appreciate all of the reviewers' and editors' helpful feedback, suggestions, and questions.

Thank you,

With kind regards!

---

## [Decision Letter · Decision Letter 1]

28 Aug 2022

PONE-D-22-05826R1The magnitude of mental distress and associated factors among a school of medicine and college of health sciences students at Debre Markos university, 2021PLOS ONE

Dear Dr. Tsegaye,

Thank you for submitting your manuscript to PLOS ONE. After careful consideration, we feel that it has merit but does not fully meet PLOS ONE’s publication criteria as it currently stands. Therefore, we invite you to submit a revised version of the manuscript that addresses the points raised during the review process.

ACADEMIC EDITOR:

As you will see in the reviews, you have done a nice job with the revision. Reviewer #3 has raised some specific and important revisions to be made.

We look forward to receiving your revised manuscript.

Kind regards,

Ali A. Weinstein, Ph.D.

Academic Editor

PLOS ONE

Journal Requirements:

Additional Editor Comments:

As you will see in the reviews, you have done a nice job with the revision. Reviewer #3 has raised some specific and important revisions to be made.

Reviewers' comments:

Reviewer's Responses to Questions

**Comments to the Author**

1. If the authors have adequately addressed your comments raised in a previous round of review and you feel that this manuscript is now acceptable for publication, you may indicate that here to bypass the “Comments to the Author” section, enter your conflict of interest statement in the “Confidential to Editor” section, and submit your "Accept" recommendation.

Reviewer #1: All comments have been addressed

Reviewer #3: All comments have been addressed

Reviewer #4: (No Response)

2. Is the manuscript technically sound, and do the data support the conclusions?

Reviewer #1: Yes

Reviewer #3: Yes

Reviewer #4: Yes

3. Has the statistical analysis been performed appropriately and rigorously? 

Reviewer #1: Yes

Reviewer #3: Yes

Reviewer #4: Yes

4. Have the authors made all data underlying the findings in their manuscript fully available?

Reviewer #1: Yes

Reviewer #3: Yes

Reviewer #4: Yes

5. Is the manuscript presented in an intelligible fashion and written in standard English?

Reviewer #1: Yes

Reviewer #3: No

Reviewer #4: Yes

6. Review Comments to the Author

Reviewer #1: The revised objectives can be included in the final paragraph of the introduction. All of the comments have been addressed adequately.

Reviewer #3: 1. General

There are too many editorial problems along the manuscript which include sentence structure, grammatical and spacing errors. There is also inconsistent use of words and phrases.

2. Abstract

• On the introduction section of the abstract, it didn't show why the study conducted

• On page 1 line 36, replace 'conducted' by 'employed'

• On the Methods part of the abstract include the study period and the total sample size.

3. Introduction

• The introduction section of this manuscript lacks coherence. You started by defining the problem then show the effect of mental health distress. The factors came later after the effect. I suggest this flow; 1st define the problem�then come across with associated factors�finally, state the effects of mental health distress.

• Page 5, line 100; don’t use the short for ‘SMCHS’ at its first mention.

4. Methods

• Page 5, line 107; enter the work “design” between the words ‘study’ and ‘was’.

• Page 5, line 109; avoid repetition on the study design.

• Your study area description lacks REFERENCE(S).

• On the exclusion criteria, is ‘March’ among the summer months in Ethiopian context? Because you stated, ‘students on summer break are excluded’

• Page 6, lines 131 & 135; replace ‘Gondar university’ by ‘University of Gondar, Ethiopia’

• On sample size determination, put the formula.

• Did you consider design effect in your sample size determination? In this regard, figure 1 is not clear and needs elaboration. Example: how many students are there for each department per each year? There are also letters like; P, M, N, E, ….., which needs clarification.

5. Results

• Page 7, line 158; remove the last sentence.

• Page 12, lines 231, 235 and 239; figures captions are misplaced. Please revise it for all.

6. Discussion

• The discussion part still needs revision. The finding is well compared with other’s work. However, the way you used to justify discrepancies of your findings with previous works is poor. Justification based on the assessment tool used, sample size difference may not be sound and convincing.

7. Conclusion and recommendation

• What you have recommended for different stakeholders based on the significant associated factors you had.

8. Figure Legend

Figure 2 & 3 are missed and Figure 6 & 7 added which are not cited in the text.

9. References

• The reference part is not written appropriately. Most of the references lack journal name, volume, number and page number (Ref. 1, 3, 5, 9, 10, 15, 20, 21, 26, 32, 33,….). There are also references without year publication. Sometimes CAPITAL letter is used (ref.10). Not only these, consider the other issues too like; use of punctuations.

10. Figures

• Figure 2 & 5 are not visible well, replace with a better visibility.

Reviewer #4: All my previous comments have been addressed. I have no further comments to add. It can be accepted for publications if they have addressed other reviewer's comments. Thanks

7. PLOS authors have the option to publish the peer review history of their article (what does this mean?). If published, this will include your full peer review and any attached files.

Reviewer #1: **Yes: **Sk Akhtar Ahmad

Reviewer #3: **Yes: **Mahmud Ahmednur

Reviewer #4: No

---

## [Author Response · Author response to Decision Letter 1]

1 Sep 2022

Author’s Point-by-Point Response to the Reviewer's and Editors Reports

The magnitude of mental distress and associated factors among A school of medicine and college of health sciences students at Debre Markos university, 2021

Corresponding Authors Dejen Tsegaye/ dejenetsegaye@gmail.com

Point by point response to Reviewers and Editors 

First and foremost, the authors would like to express their gratitude to the PLOSE ONE Journal editors and reviewers for thoroughly evaluating this work and offering the required corrections. We made changes based on the feedback we received and presented each comment point by point. The authors attempted to address all of the concerns expressed by the editorial board and reviewers. Please note that the response was written in blue font.

Authors' responses to the editors' remarks

Editor: As you will see in the reviews, you have done a nice job with the revision. Reviewer #3 has raised some specific and important revisions to be made.

Response: Thank you! Reviewer three provided us with some very significant feedback, and we made an effort to make the necessary corrections.

Authors' responses to the authors' remarks

REVIEWER #3

Comment: 1. General: 

There are too many editorial problems along the manuscript which include sentence structure, grammatical and spacing errors. There is also inconsistent use of words and phrases.

Response: The authors made an effort to review the paper and make corrections for grammatical, spacing, and other concerns, such as terms or phrases that were used inconsistently in relation to your feedback.

Comment: 2. Abstract 

• On the introduction section of the abstract, it didn't show why the study conducted

Response: I appreciate your advice. We have inserted a clause that clarifies the aim of the investigation.

• On page 1 line 36, replace 'conducted' by 'employed'

Response: Thank you for your advice. We made the necessary changes.

• On the Methods part of the abstract include the study period and the total sample size.

Thank you for your advice. We have added the study period and total sample size.

Comment: 3. Introduction

• The introduction section of this manuscript lacks coherence. You started by defining the problem then show the effect of mental health distress. The factors came later after the effect. I suggest this flow; 1st define the problem�then come across with associated factors�finally, state the effects of mental health distress.

Response: Thank you for your advice. We made the necessary changes.

• Page 5, line 100; don’t use the short for ‘SMCHS’ at its first mention.

Response: Thank you for your advice. We made the necessary changes.

Comment: 4. Methods

• Page 5, line 107; enter the work “design” between the words ‘study’ and ‘was’.

Response: Thank you for your advice. We made the necessary changes.

• Page 5, line 109; avoid repetition on the study design.

Response: Thank you for your input; we've made a change as a result of your idea.

• Your study area description lacks REFERENCE(S).

Response: Thank you for your suggestion. We have added a reference.

• On the exclusion criteria, is ‘March’ among the summer months in Ethiopian context? Because you stated, ‘students on summer break are excluded’

Response: When viewed in the context of Ethiopia, March is not summer. However, starting March, departments from the school and college are permitted to leave the university for break early if they were able to complete all of their courses. This university's curriculum is year-based rather than semester-based. I appreciate you spending the time to make such a thoughtful response.

• Page 6, lines 131 & 135; replace ‘Gondar university’ by ‘University of Gondar, Ethiopia’

Response: Thank you for your input; we've made a change as a result of your idea.

• On sample size determination, put the formula.

Response: We have added the formula based on your suggestion. Thank you.

• Did you consider design effect in your sample size determination? In this regard, figure 1 is not clear and needs elaboration. Example: how many students are there for each department per each year? There are also letters like; P, M, N, E, ….., which needs clarification.

Response: The study was unable to take design effect into account. The design effect can be utilized to correct the estimated sampling variance, as is well knowledge. The study took into account/involved all departments/sections, hence a design effect is not required. The stratification and proportional distribution of sample size to each department were simply depicted in the picture. The departments were revealed by the personalities that were referenced there. Its legend is supplied below the figure. I appreciate you spending the time to make such a thoughtful response.

Comment: 5. Results

• Page 7, line 158; remove the last sentence.

Response: Thank you for your advice. We made the necessary changes.

• Page 12, lines 231, 235 and 239; figures captions are misplaced. Please revise it for all.

Response: Thank you. We revised and made the necessary change.

Comment: 6. Discussion

• The discussion part still needs revision. The finding is well compared with other’s work. However, the way you used to justify discrepancies of your findings with previous works is poor. Justification based on the assessment tool used, sample size difference may not be sound and convincing.

Response: Thank you for your advice. We tried to make necessary modifications.

Comment: 7. Conclusion and recommendation

• What you have recommended for different stakeholders based on the significant associated factors you had.

Response: Based on the finding, recommendations are given to the concerned body. Thank you!

Comment: 8. Figure Legend

Figure 2 & 3 are missed and Figure 6 & 7 added which are not cited in the text.

Response: For this feedback, we've made an effort to carefully go over the entire document. I appreciate you spending the time to make such a thoughtful response.

Comment: 9. References

• The reference part is not written appropriately. Most of the references lack journal name, volume, number and page number (Ref. 1, 3, 5, 9, 10, 15, 20, 21, 26, 32, 33,….). There are also references without year publication. Sometimes CAPITAL letter is used (ref.10). Not only these, consider the other issues too like; use of punctuations.

Response: For citation, we used the endnote program; manual referencing was not used. Because of this, references are immediately cited from the software without any modification, which accounts for all of the improper referencing methods discussed above. Now, in response to your comment, we have made the necessary changes. I appreciate your thoughts.

Comment: 10. Figures

• Figure 2 & 5 are not visible well, replace with a better visibility.

Response: Thank you. To improve visibility, we created another figure for each. The visibility has improved recently.

We appreciate all of the reviewers' and editors' helpful feedback, suggestions, and questions.

Thank you,

With kind regards!

---

## [Decision Letter · Decision Letter 2]

12 Sep 2022

The magnitude of mental distress and associated factors among a school of medicine and college of health sciences students at Debre Markos university, 2021

PONE-D-22-05826R2

Dear Dr. Tsegaye,

We’re pleased to inform you that your manuscript has been judged scientifically suitable for publication and will be formally accepted for publication once it meets all outstanding technical requirements.

Kind regards,

Ali A. Weinstein, Ph.D.

Academic Editor

PLOS ONE

Additional Editor Comments:

Please see the final reviewer comments below. For the final version of the paper, you may want to revise/recheck these.

Reviewers' comments:

Reviewer's Responses to Questions

**Comments to the Author**

1. If the authors have adequately addressed your comments raised in a previous round of review and you feel that this manuscript is now acceptable for publication, you may indicate that here to bypass the “Comments to the Author” section, enter your conflict of interest statement in the “Confidential to Editor” section, and submit your "Accept" recommendation.

Reviewer #3: All comments have been addressed

2. Is the manuscript technically sound, and do the data support the conclusions?

Reviewer #3: Yes

3. Has the statistical analysis been performed appropriately and rigorously? 

Reviewer #3: Yes

4. Have the authors made all data underlying the findings in their manuscript fully available?

Reviewer #3: Yes

5. Is the manuscript presented in an intelligible fashion and written in standard English?

Reviewer #3: Yes

6. Review Comments to the Author

Reviewer #3: Please check the sample size determination farmula. The other thing still on the discussion on page 16, line 292-303, the first three paragraphs are not neccessary. On the references still it needs revision.

7. PLOS authors have the option to publish the peer review history of their article (what does this mean?). If published, this will include your full peer review and any attached files.

Reviewer #3: **Yes: **Mahmud Ahmednur Mohammed

---

## [Editor Report · Acceptance letter]

15 Sep 2022

PONE-D-22-05826R2 

The magnitude of mental distress and associated factors among a school of medicine and college of health sciences students at Debre Markos university, 2021 

Dear Dr. Tsegaye:

I'm pleased to inform you that your manuscript has been deemed suitable for publication in PLOS ONE. Congratulations! Your manuscript is now with our production department. 

Kind regards, 

on behalf of

Dr. Ali A. Weinstein 

Academic Editor

PLOS ONE